# Examining the Public Messaging on ‘Loneliness’ over Social Media: An Unsupervised Machine Learning Analysis of Twitter Posts over the Past Decade

**DOI:** 10.3390/healthcare11101485

**Published:** 2023-05-19

**Authors:** Qin Xiang Ng, Dawn Yi Xin Lee, Chun En Yau, Yu Liang Lim, Clara Xinyi Ng, Tau Ming Liew

**Affiliations:** 1Health Services Research Unit, Singapore General Hospital, Singapore 169608, Singapore; ng.qin.xiang@u.nus.edu; 2Ministry of Health Holdings Pte Ltd., Singapore 099253, Singapore; 3School of Medicine, Dentistry and Nursing, University of Glasgow, Glasgow G12 8QQ, UK; 4NUS Yong Loo Lin School of Medicine, Singapore 117597, Singapore; 5Department of Psychiatry, Singapore General Hospital, Singapore 169608, Singapore; 6SingHealth Duke-NUS Medicine Academic Clinical Programme, Duke-NUS Medical School, Singapore 169857, Singapore; 7Saw Swee Hock School of Public Health, National University of Singapore, Singapore 117549, Singapore

**Keywords:** public messaging, lonely, social media, machine learning, natural language processing, thematic analysis

## Abstract

Loneliness is an issue of public health significance. Longitudinal studies indicate that feelings of loneliness are prevalent and were exacerbated by the Coronavirus Disease 2019 (COVID-19) pandemic. With the advent of new media, more people are turning to social media platforms such as Twitter and Reddit as well as online forums, e.g., loneliness forums, to seek advice and solace regarding their health and well-being. The present study therefore aimed to investigate the public messaging on loneliness via an unsupervised machine learning analysis of posts made by organisations on Twitter. We specifically examined tweets put out by organisations (companies, agencies or common interest groups) as the public may view them as more credible information as opposed to individual opinions. A total of 68,345 unique tweets in English were posted by organisations on Twitter from 1 January 2012 to 1 September 2022. These tweets were extracted and analysed using unsupervised machine learning approaches. BERTopic, a topic modelling technique that leverages state-of-the-art natural language processing, was applied to generate interpretable topics around the public messaging of loneliness and highlight the key words in the topic descriptions. The topics and topic labels were then reviewed independently by all study investigators for thematic analysis. Four key themes were uncovered, namely, the experience of loneliness, people who experience loneliness, what exacerbates loneliness and what could alleviate loneliness. Notably, a significant proportion of the tweets centred on the impact of the COVID-19 pandemic on loneliness. While current online interactions are largely descriptive of the complex and multifaceted problem of loneliness, more targeted prosocial messaging appears to be lacking to combat the causes of loneliness brought up in public messaging.

## 1. Introduction

Humans are inherently social creatures, and loneliness is a complex human emotion, broadly defined as a perceived state of solitude or lack of meaningful relationships with others [1]. Loneliness is thought to affect around 30% of older adults globally [2]. A growing body of longitudinal research also indicates that feelings of loneliness are prevalent and were exacerbated by the Coronavirus Disease 2019 (COVID-19) pandemic [3]. This is of public health significance as multiple studies have found loneliness to be independently associated with increased all-cause mortality, even after controlling for age, sex, chronic diseases, alcohol use, smoking, self-rated health and functional status [4,5]. Risk factors of loneliness include social isolation, institutionalization and a low socioeconomic status [6].

Today, with the widespread use and availability of the internet and social media platforms such as Twitter, people are increasingly turning to these platforms not only for general information seeking and sharing [7] and medical information dissemination [8] but also for emotional support. This is especially true for individuals who experience loneliness, who may seek solace and advice from online forums and social media platforms dedicated to discussing this issue [9,10,11]. Individuals may post about their experiences, and previous studies have characterised the public discourse surrounding loneliness [12] and described how users respond to individuals who express loneliness on Twitter [13]. However, there is a paucity of research specifically examining the public messaging on loneliness by organisations on social media, which may have a far-reaching influence.

Anchored in a transdisciplinary approach, the present study therefore aimed to investigate the public messaging on loneliness via an unsupervised machine learning analysis of posts made by organisations on Twitter. We specifically examined tweets put out by organisations (referring to companies, agencies or common interest groups), as the public may view them as more credible information as opposed to subjective individual opinions. In an earlier study that looked at the public emotional diffusion over COVID-19-related tweets posted by health agencies in the United States (US), it was found that the tweets posted by the US Centers for Disease Control and Prevention (CDC) and the Food and Drug Administration (FDA) had profound influence on the public [14]. As the use of social media is increasingly pervasive, characterising the public messaging related to loneliness over Twitter and exploring the issues raised herein may better inform online loneliness messaging and interventions.

## 2. Methods

The methodology for the present study was adapted from previous published works leveraging machine learning approaches [15,16]. Using Twitter as the social media platform of choice and the search term ‘#loneliness’, we extracted tweets that were posted in English from 1 January 2012 to 1 September 2022. The tweets were extracted from Twitter’s Application Programming Interface (API) platform, using an academic developer account, which allows downloads of all tweets (i.e., not sampling) of up to 10 million tweets per month. Retweets and duplicate tweets (i.e., tweets with identical sentences and words) were excluded from this study. There was no restriction in the country of origin of the tweets.

Focusing on the public messaging on loneliness, only tweets by organisations were included in this study, as selected using BERT Named Entity Recognition (NER), which locates and identifies named entities, e.g., person names and organisations mentioned in unstructured free text [17]. BERT NER has been trained using a pre-training and fine-tuning approach, and it uses a sequence labelling approach, where the model takes a sequence of tokens (words or sub-words) as input and predicts a label for each token, indicating whether it belongs to an entity or not, and if so, what type of entity it is. It is able to recognise four types of entities: location (LOC), organisations (ORG), person (PER) and miscellaneous (MISC). 

BERTopic, a topic modelling technique that leverages state-of-the-art Bidirectional Encoder Representations from Transformers (BERT) embeddings and class-based term frequency–inverse document frequency (TF–IDF) [18], was then applied to generate dense clusters of interpretable topics around the public messaging on loneliness and highlight the key words in the topic descriptions. BERTopic was chosen over other techniques such as Latent Dirichlet Allocation (LDA) as it is based on BERT embeddings, which are contextualised word representations learned through pre-training on large amounts of text data [19]. This means that BERTopic can take into account the context and meaning of words and provide more accurate and meaningful topic representations as compared to LDA, which uses a bag-of-words approach [19,20]. In BERTopic, the following hyperparameters were used: ngram of up to three words, minimum topic size of 0.5% of the total tweets, embedding model of “all-mpnet-base-v2”. Unlike supervised machine learning, in unsupervised machine learning, the data does not have a known outcome, and as a language model, BERTopic has been pre-trained on a large volume of free text and has demonstrated good stability across domains [19]. For each topic, we also classified the tweets belonging to these topics as either positive or negative sentiment using SiEBERT, a pre-trained sentiment in English analysis model [21].

We also calculated the mean public attention scores for the negative and positive tweets under each topic; the public attention score for each tweet was calculated as the sum of the retweet count, reply count, like count and quote count. The measure of public attention was expressed as a numerical score, as adapted from previous public opinion research [22]. Where applicable, R (version 3.6.3) and Python (version 3.7.13) were used for all quantitative analyses.

The topics and topic labels were then reviewed independently by all study investigators for inductive and iterative thematic analysis, as guided by Braun and Clarke [23]. Thematic analysis was chosen to enrich the analysis, as it is an adaptable method that can be applied to a wide range of qualitative data and topics of interest, including tweets. It does not rely on pre-determined categories or frameworks, allowing for an open and iterative process of data analysis [24]. The study investigators began by familiarising themselves with the keywords and sample tweets, produced preliminary codes, formulated overarching themes, reviewed and refined themes, defined and specified themes and produced a write-up. Coding disagreements were resolved through discussion among the study investigators until a consensus was reached.

Ethics approval for the study was granted by the SingHealth Centralised Institutional Review Board (CIRB) of Singapore (reference number: 2021/2717). No human participants were directly involved in the study. All data in the present study were collected according to Twitter’s terms of use [25].

## 3. Results

A combined total of 287,260 tweets were extracted. Of these, there were 212,558 unique tweets in English. After removing tweets by individual Twitter users, a total of 68,345 tweets posted by organisations over Twitter were analysed. The tweets’ extraction and selection process are summarized in Figure 1.

The public messaging on loneliness appeared to centre around 12 topics based on topic modelling (Table 1). These topics could be further grouped under four main themes based on qualitative thematic analysis by the study team. The themes reflected the different facets around the concept of loneliness, namely, 1. the experience of loneliness, 2. people who experience loneliness, 3. what exacerbates loneliness and 4. what could alleviate loneliness. 

The topics comprised both tweets with positive and negative sentiments. A greater proportion of the tweets framed the public messaging on loneliness with a positive sentiment. However, based on the mean public attention scores, it can be inferred that both tweets of positive and negative sentiments drew comparable amounts of public attention from Twitter users.

We also analysed the temporal variations in the number of tweets belonging to each Theme (Figure 2), which revealed an uptick in the public messaging about people who experience loneliness (Theme 2) from 2018 onwards and a significant increase in the public messaging about what exacerbates loneliness (Theme 3) from 2018 onwards, with significant peaks around the year 2019 and 2020, suggesting possible concurrences with the ongoing COVID-19 pandemic, although the frequencies declined in 2022.

## 4. Discussion

Based on our study findings, analysis of the corpus of tweets by organisations on Twitter pertaining to loneliness revealed four key themes: namely, the experience of loneliness, people who experience loneliness, what exacerbates loneliness and what could alleviate loneliness. In general, research has emphasised the complex and multifaceted nature of loneliness, highlighting the interactions between specific human behaviours, emotions (feeling unloved or unwanted) and thoughts of negative and self-depreciating nature [26,27]. Loneliness can result from a variety of factors, including social and environmental conditions, individual differences in personality and attachment style and life events such as loss or transitions. It may also arise from other cognitive aspects (e.g., the perceived discrepancy between desired and actual social relationships) and have multiple dimensions (social, emotional and existential) [28].

The public messaging on loneliness broadly reflects these ideas, and a commonality is the focus on experiences of social distress and its effects on one’s thoughts (e.g., self-depreciation) and behaviour (e.g., social isolation). It is well known that socioeconomic and situational variables may contribute to feelings of loneliness, and it is the perception of being alone which may make the individual feel lonely. This public messaging also suggests possible methods to prevent and alleviate feelings of loneliness, e.g., religiousness and human–animal interactions. Social prescribing is an emerging intervention for loneliness [29]. Individuals probably seek emotional and informational supports, and online interventions may be aimed at providing these.

Notably, a significant proportion of the tweets were directed at the impact of the COVID-19 pandemic on loneliness (Topic 2), which had the largest number of tweet mentions only after the general topic of social isolation and loneliness (Topic 1). We also saw an increase in the number of tweets on what exacerbates loneliness (Theme 3) from 2018 onwards. This suggests that COVID-19 was a possible contributing factor to global levels of loneliness, although this theme has been increasing since 2018 while the COVID-19 pandemic only reached pandemic status in 2020, and the frequencies of the tweets also came down in 2022 despite the ongoing pandemic. Nonetheless, research has shown a deterioration in mental health of the general population [30] and an increase in perceived levels of loneliness during the pandemic, especially with social distancing measures in place [31]. Yet, there is a paucity of research regarding loneliness interventions related to the pandemic [32]. Further research could look into interventions implemented during the pandemic or targeted at the legacy of COVID-19 and its longitudinal effects on loneliness.

A significant proportion of the tweets also highlighted the problem of loneliness faced by the elderly population, “Loneliness is the epidemic of the old” and “Bone, joint and muscle conditions are a big issue for people. They’re debilitating; life-restricting; can result in isolation and #loneliness”, and those living with dementia as well as their caregivers, “The loneliness of #dementia #loneliness”. This also highlights the need to socially re-engage, “The Less I Go Out, the More Afraid I Am to Go Out. Why?”

In terms of the public messaging by organisations, there were no glaring misinformation or misconceptions. While current online interactions are largely descriptive of the problem of loneliness, more targeted prosocial messaging appears to be lacking to combat the causes of loneliness brought up in public messaging. This is a missed opportunity as the public may view posts by organisations as being more credible. Moreover, research suggests that engaging in prosocial behaviour may help overcome one’s feelings of loneliness [33], and this recommendation could be easily encapsulated in a tweet. Technology may help promote social connection and maintain social connectedness as public health messaging. Simple messages to prioritise self-care activities, to reach out to friends and family, to practice gratitude and information on tools and strategies to help manage one’s emotions and improve social connections could be easily weaved into public messaging. It is also essential to encourage individuals to seek professional support if loneliness is affecting their mental health and well-being.

Nonetheless, our study findings should be interpreted in light of the following shortcomings. First, Twitter was the only social media platform analysed and may not be representative of all social media users, and moreover, the majority of Twitter users (and organisations) on Twitter are from North America, Europe and the United Kingdom, and as such, the findings are not representative of the general public. Second, although we used machine learning approaches with good face validity to analyse the sentiment of these tweets, the automated algorithm could have erroneously categorised tweets with more nuanced or complex emotions. For example, “Any good examples out there of UK universities addressing the potential for #loneliness among their socially isolated researchers working from home?” was categorised as a tweet showing negative sentiment. Such misclassifications could have skewed our results. Third, the organisations examined in the present study were not compared in terms of their nature of business, scale or size, which could also affect the reach and effectiveness of their messaging. Fourth, we cannot rule out the presence of bot accounts on Twitter that may have a name similar or identical to that of an organisation, although research has shown that such bot accounts make up only a small proportion of users and tweets on the platform [34]. The sentiment expressed by these bot accounts is still relevant, however, as it may influence what people are potentially exposed to on the platform.

## 5. Conclusions

In conclusion, the present study characterised the public messaging on loneliness posted by organisations on Twitter. Collectively, the tweets reflect the multidimensional and multifaceted nature of the concept of loneliness. This could be due to a lack of social engagement; to situational variables, e.g., holidays may contribute to feelings of loneliness; and to the fact that loneliness is more prevalent as one ages. There are potential avenues for public health intervention, and public messaging could offer more motivation and encouragement around lifestyle and social prescribing for loneliness.

## Figures and Tables

**Figure 1 healthcare-11-01485-f001:**
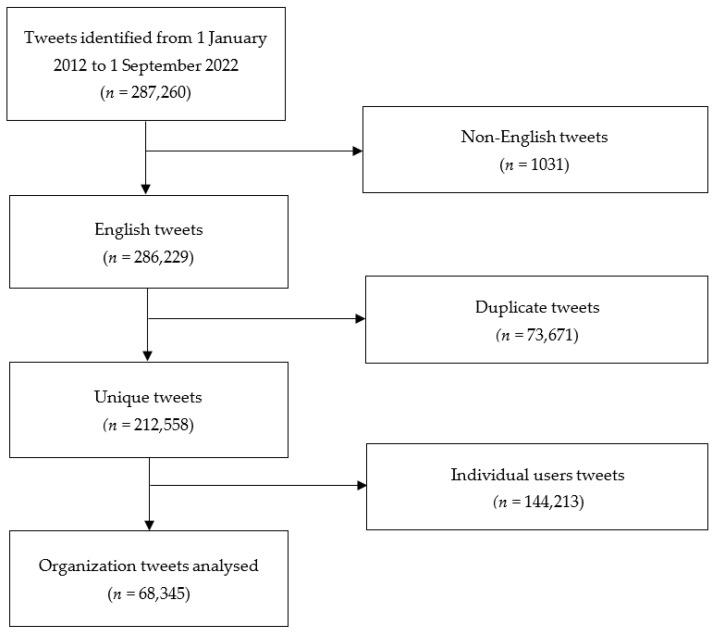
Flowchart showing the tweet selection process.

**Figure 2 healthcare-11-01485-f002:**
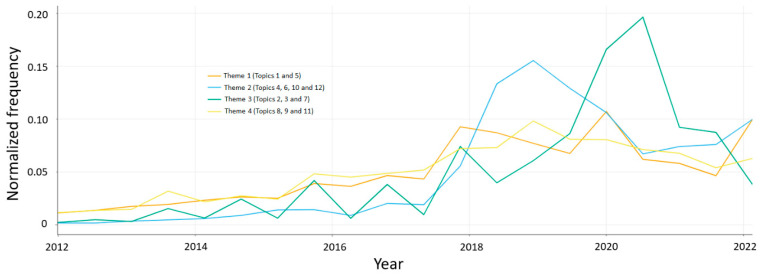
Temporal trends in the normalised frequency of tweets belonging to Theme 1 (Topics 1 and 5), Theme 2 (Topics 4, 6, 10 and 12), Theme 3 (Topics 2, 3 and 7) and Theme 4 (Topics 8, 9 and 11).

**Table 1 healthcare-11-01485-t001:** Themes related to the public messaging on loneliness, with the respective topic labels and sample tweets.

Theme, Topic Labels *(Keywords)*	Sample Tweets	Number of Tweets, n (%)	Public Attention Score, Mean (SD) ^1^
**Theme 1: What is loneliness**			
Topic 1: Social isolation and loneliness (*older*, *tackle*, *mental*, *loneliness loneliness*, *mentalhealthawarenessweek*, *mental health*, *tackle loneliness*, *combat*, *socialisolation*, *isolation loneliness*)	Positive sentiment “On this #ForestMartyrsDay want to draw your attention toward thousands of #forest personals serving in remote corners of country. In creeks, on hills & cores unnoticed & unrecognized. Fighting all kind of #crimes, #diseases & #loneliness. Scores laying lives. Do remember them.” “However lonely we feel sometimes, we are not alone. Millions of people around the world are feeling a similar way. Let s tackle loneliness and not let anybody feel as if they don t matter. #MentalHealthAwarenessWeek #Loneliness #IveBeenThere”	36,335 (53.16)	4.90 (54.78)
	Negative sentiment “#LONELINESS Is: No Emails In The Inbox… No Friends On Facebook… No Retweets By Anyone… No Comments On The Blog...” “Caught up in a middle of something, wishing to talk to someone and you can’t find one, what a bad day #loneliness.”	6171 (9.03)	2.74 (55.10)
Topic 5: Loneliness and sadness *(loneliness validation*, *lettinggo*, *sadness loneliness validation*, *friendship sadness loneliness)*	Positive sentiment “Free your mind from #Anxiety, #Fear, #Stress, #Loneliness & #Loss; take a step back, #Focus & #Breathe & embrace #Happiness, #Peace, #Tranquility, #Harmony & #InnerPeace. ‘Breathe’ “What ever this man says believe… I love myseld do you?? #KeanuReeves #ThursdayThoughts #ThursdayMotivation #thursdayvibes #quote #quotesoftheday #lonely #loneliness #LifeLessons #lifegoals #goodmorning #goodmorningyall”	1385 (2.03)	0.32 (2.06)
	Negative sentiment “I don t want to remember good memories, cus there s much more bad memories. #sad #sadness #depression #depressed #hurt #Broken #loneliness #lonely #tired” “The end of the world #end #world #contamination #endless #trash #time #loneliness #boy”	50 (0.07)	1.52 (4.12)
**Theme 2: People who experience loneliness**			
Topic 4: Loneliness and ageing *(ageing*, *harpenden*, *health homecare*, *harpenden stalbans)*	Positive sentiment “Health Benefits of Being a Grandparent #seniors #grandparenting #Grandma #grandpa #health #heathy #wellness #loneliness” “Bone, joint and muscle conditions are a big issue for people. They’re debilitating; life-restricting; can result in isolation and #loneliness, often an #invisibleillness. Isnt it time we tackled the elephant in the room? Support by using #BoneJointWeek”	1338 (1.96)	0.48 (2.12)
	Negative sentiment “Confusion, financial pressure, discomfort: older people can struggle with sustainable living, despite its obvious benefits #nans4grans #olderpeople #ageing #seniors #Harpenden #StAlbans #Health #Homecare #loneliness #respitecare” “Just a teaser re #Compassionate Ageism blog for: * The #loneliness of the long distance phone call * Health & Social Care Conveyor belt *Carrots &Coffins via #TheCoopWay * Cash cow Age Sector false assumptions *Boomer’s bloomers… just for starters…”	483 (0.71)	0.16 (0.80)
Topic 6: Loneliness at workplace *(workplace*, *loneliness workplace*, *employees*, *business)*	Positive sentiment “#Loneliness cannot end by changing company. It can only end when you discover your real nature for yourself.—#Gurudev” “How and where we work is changing, but the importance of your relationships with coworkers is not. Attend Connect+Work on July 29 @ 2pm ET to learn how to avoid #loneliness and deepen connections with your team.”	1000 (1.46)	2.54 (5.56)
	Negative sentiment “Work-from-home blues are real. #RemoteWork #Freelancer #WorkFromHome #Burnout #Loneliness” “Workers are lonelier than ever before resulting in a low-performing, disloyal, and #burntout workforce. #leadership #connection #loneliness”	195 (0.29)	1.47 (4.12)
Topic 10: Loneliness and social anxiety *(socialanxiety*, *eft*, *socialphobia*, *selfhelp loneliness)*	Positive sentiment “Overcome Fear & Worry via #fear #worry #anxiety #panic #depression #loneliness” “Witness how you will find relief using the EFT Technique. Go to the link here #socialanxiety #anxiety #shame #selfhelp #loneliness #fearofrejection #eft #socialanxietyproblems #socialphobia #selfacceptance #mentalhealth #anticipatoryanxiety”	588 (0.86)	0.63 (1.28)
	Negative sentiment “The Less I Go Out, the More Afraid I Am to Go Out. Why? #FightLoneliness #Nottingham #Loneliness #Lonely #LonelyTogether #TheWolfpackProject #Help #SocialIsolation #Lockdown #goout” “My social anxiety disorder level—it is so hard to speak with you Alexa, I’m so sorry. #socialanxietydisorder #sad #Alexa #witch #witches #witchcraft #depression #anxiety #social #wicca #wiccan #pagan #Emotions #GirlsOfTwitter #level #lonelyplanet #loneliness #lonely”	21 (0.03)	1.33 (2.89)
Topic 12: Loneliness and dementia *(dementia*, *dementia loneliness*, *loneliness dementia*, *alzheimers)*	Positive sentiment “81% of carers have felt lonely or isolated as a result of looking after a loved one* Caring for someone with #dementia can be isolating. This #LonelinessAwarenessWeek, we’re sharing our tips on some of the ways you can combat #loneliness… *statistic from Carers UK” “How one man combatted the #loneliness of dementia by becoming a campaigner #charitytuesday”	368 (0.54)	5.21 (8.72)
	Negative sentiment “The loneliness of #dementia #loneliness” “’Loneliness is the epidemic of the old’ Yesterday, today & tomorrow are the same without COMPANY. #loneliness #dementia”	47 (0.07)	2.70 (5.78)
**Theme 3: What exacerbates loneliness**			
Topic 2: Loneliness during the COVID-19 pandemic *(pandemic*, *covid19*, *covid*, *covid 19)*	Positive sentiment “#Covid19 will have a huge impact on #loneliness and those who are already lonely may be more vulnerable than ever. Join our #TacklingLoneliness Twitter chat this Thursday with our Loneliness Lead & from for an important discussion.” “Over the next few weeks and months during #COVID19, we will be sharing what works to improve social connection, #community wellbeing, and reduce #loneliness—because evidence shows #work, #health & relationships are vital for #wellbeing. #CommunityQuarantine #SocialDistance”	2199 (3.22)	4.54 (9.40)
	Negative sentiment “39% of people who say they are always or often lonely feel as though they can’ t cope with the pandemic. See new new report on #loneliness” “Many living w/#InvisibleDisabilities have battled #isolation & #loneliness before #Covid & will continue! Both are detrimental to our #Health & well-being! Will people have more compassion now that they have experienced #SocialDistancing/#lockdowns?“	405 (0.59)	2.11 (4.64)
Topic 3: Loneliness during the holidays *(christmas*, *holidays*, *loneliness christmas*, *year)*	Positive sentiment “Officers responded to a concern for welfare call today … found a lonely woman in her 80s. The officers helped wash up, change her light bulbs & have made appropriate referrals. One of the officers is even going to visit her on Christmas Day. #Lonely_Christmas #loneliness“ “For so many reasons winter can be the toughest and loneliest of all seasons for somebody on their own, but there are always ways we can communicate and reconnect to show them they are not alone. #Loneliness #allontheboard ’ Board on display in North Greenwich’.”	2270 (3.32)	5.37 (51.22)
	Negative sentiment “Anyone else feeling extra lonely this holiday cause of covid/isolation/holidays in general? #holidays2020 #loneliness” “No one should spend #Christmas alone, but people with a #learningdisability may feel excluded during the festive season #LD #loneliness”	204 (0.3)	2.49 (6.16)
Topic 7: Loneliness and social media *(technology*, *media*, *social media*, *socialmedia)*	Positive sentiment “‘The goal is not to increase #technology addiction but to use it as a force for positive change.’ My latest piece about how #InclusiveDesign can fight the global #loneliness epidemic #WritingCommunity ty editors &” “Spending too much time on social media has repeatedly been linked to #depression and #loneliness. This should be required reading for every parent and educator. Some important insights on the impact of #technology: #UndercoverHigh”	848 (1.24)	3.20 (5.79)
	Negative sentiment “Technology has made loneliness much worse. If we are not in constant communication with someone, we feel terribly alone. #loneliness ” “As we expect more from technology, do we expect less from each other? #relationships #friendship #loneliness	282 (0.41)	1.75 (3.79)
**Theme 4: How to address loneliness**			
Topic 8: Loneliness and religion *(god*, *jesus*, *church*, *faith)*	Positive sentiment “What is spiritual bypassing? gives us the scoop at #AoC: #loneliness #depression #alone #love #sadness” “How do you deal with loneliness? Mark sits down with Donald Pinnock, a Minister of the Gospel in the Church Of Christ, to find a solution to his problem. Catch a new episode of The Solution on #loneliness #LetsTalk #BibleStudy #TheSolution #IglesiaNiCristo”	941 (1.38)	2.76 (6.99)
	Negative sentiment “Church minister reflects on leading paupers funerals with no mourners present #funeralpoverty #loneliness” “Abuse of power often leads to #socialthinning Eg loss of church community, loss of spouse, loss of family and friends. Clique churches lead to more isolation and #loneliness not accepting those different. #survivorpain”	92 (0.13)	50.08 (471.78)
Topic 9: Loneliness and pets *(pets*, *pet*, *dog*, *dogs)*	Positive sentiment “It may only be August but we re already on the hunt for our Christmas #CatventCalendar #cats! Has your cat improved your #MentalHealth, helped combat #loneliness or just made your house feel like home this year? Tweet us a photo and your cat s story!” “Research suggests that human-animal interaction may benefit the #mentalhealth of people of all ages. Here’s how we’re supporting studies to find out how interacting with #therapydogs may help ease #loneliness and social isolation especially in older adults”	590 (0.86)	6.90 (24.90)
	Negative sentiment “The true price of veal and of supporting industries who treat their animals like trash. #sadness #loneliness #modernanimalagriculture” “Saddening results survey on number of older people separated from their much loved pet #loneliness”	61 (0.09)	4.92 (17.30)
Topic 11: Loneliness as portrayed in media *(film*, *short film*, *films*, *video loneliness)*	Positive sentiment “Loneliness is the story of far too many older people. It’s crippling, can be life-limiting and it’s time to change this. And we can. Together. This moving film, ‘Please Answer’, was written, directed (and has a cameo by) #last1000days #loneliness” “‘A Simple Chat’—by PC 1942 A touch of basic human kindness can make ALL the difference. Talk. Listen. Talk some more. #mentalhealth #SuicidePrevention #SuicideAwareness #depression #MentalHealthAwarenessWeek #loneliness #KindnessMatters #ASimpleChat”	539 (0.79)	7.40 (49.85)
	Negative sentiment “Video: The age of #loneliness” “To Abridge my 30 min sob story on YouTube, I try to use #livestreaming as means to battle #depression and #loneliness. When I have streams where nobody’s around, I feel the purpose is defeated. Partially my fault that I don’ t know how to keep talking when no one is talking....”	36 (0.05)	2.06 (4.22)

^1^ Public attention score for each tweet was calculated as the sum of the retweet count, reply count, like count and quote count.

## Data Availability

The datasets generated during and/or analysed during the current study are available from the corresponding author upon reasonable request.

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
