# Peer review of "Examining the Public Messaging on ‘Loneliness’ over Social Media: An Unsupervised Machine Learning Analysis of Twitter Posts over the Past Decade"

_healthcare, 2023, doi:10.3390/healthcare11101485_

Round 1

Reviewer 1 Report

Details of how tweets were collected from Twitter (API, scraping?) should be provided. This will indicate whether *all* tweets with #loneliness was included in the study or only a sample.

It is not clear how the distinction of an "organization" and "person" was made. Line 80 seems to indicate both persons and organisations were included ("person names and organisations mentioned in 80 unstructured free-text").

Any hyperparameters used in BERTopic should be provided. It is not clearly stated how many topics were discovered by BERTopic (12?). Were there topics that were discovered by BERTopic that were not considered relevant by the researchers?

The conclusion that "COVID-19 was likely a contributing factor to global levels of 181 loneliness" is not strong considering that this theme has been increasing since 2018 while the COVID pandemic reached pandemic status only in 2020.

Author Response

To begin, we thank the reviewer for taking the time and effort to review our manuscript.

Comment 1: Details of how tweets were collected from Twitter (API, scraping?) should be provided. This will indicate whether *all* tweets with #loneliness was included in the study or only a sample.

Reply 1: Thank you for the comment. The tweets were extracted from Twitter’s Application Programming Interface (API) platform, using an academic developer account, which allows downloads of all tweets (i.e. not sampling) of up to 10 million tweets per month. We have now added this information to our Methods section.

Comment 2: It is not clear how the distinction of an "organization" and "person" was made. Line 80 seems to indicate both persons and organisations were included ("person names and organisations mentioned in 80 unstructured free-text").

Reply 2: We apologise for any confused caused. Tweets by organisations were selected using BERT Named Entity Recognition (NER), which locates and identifies named entities, e.g., person names and organisations mentioned in unstructured free-text. BERT NER has been trained using a pre-training and fine-tuning approach, and it uses a sequence labeling approach, where the model takes a sequence of tokens (words or subwords) as input and predicts a label for each token indicating whether it belongs to an entity or not, and if so, what type of entity it is. It is able to recognize four types of entities: location (LOC), organizations (ORG), person (PER) and Miscellaneous (MISC). We have now provided further elaboration and explanation in the Methods section.

Comment 3: Any hyperparameters used in BERTopic should be provided. It is not clearly stated how many topics were discovered by BERTopic (12?). Were there topics that were discovered by BERTopic that were not considered relevant by the researchers?

Reply 3: Thank you for the comment. In BERTopic, the following hyperparameters were used: ngram of up to 3 words, minimum topic size of 0.5% of the total tweets, embedding model of “all-mpnet-base-v2”. We have now added this information in the Methods section. 

Comment 4: The conclusion that "COVID-19 was likely a contributing factor to global levels of 181 loneliness" is not strong considering that this theme has been increasing since 2018 while the COVID pandemic reached pandemic status only in 2020.

Reply 4: Thank you for the comment. We agree with the reviewer and have tempered our discussion, "This suggest that COVID-19 was a possible contributing factor to global levels of loneliness although this theme has been increasing since 2018 while the COVID-19 pandemic only reached pandemic status in 2020, and the frequencies of the tweets have also come down in 2022 despite the ongoing pandemic."

Reviewer 2 Report

The work presented in this paper aims to investigate the public messaging on loneliness via an unsupervised machine-learning analysis of posts by organizations on Twitter. The authors examined tweets from organizations (companies, agencies, or common interest groups). A total of 68,345 unique tweets in English posted by organizations on Twitter were extracted and analyzed in this work. The work seems novel. However, parts of the paper need improvement. The following represents my comments/feedback for the improvement of different parts of this paper:

1.       Rewrite the Methods section to clearly highlight how the Tweets were extracted. Was the Standard Search API used or the Advanced Search API? How long did the data collection take? How did the authors design their data collection to comply with the rate limits of accessing the Twitter API?

2.       Why was BERT Named Entity Recognition (NER) used instead of other topic modeling approaches such as LDA?

3.       Please elaborate on how Twitter usernames were analyzed to deduce those were specific organizations. How accurate was this approach? How did the authors take into consideration that there may be bot accounts on Twitter which could have a name similar/identical to an organization?

4.       The second paragraph in the Introduction section outlines just a couple of uses of social media networks such as Twitter in the context of COVID-19. Please elaborate on why people used Twitter in this context with supporting references – for instance: general information seeking and sharing (https://doi.org/10.3390/covid2080076), medical information dissemination (https://doi.org/10.1017/cem.2020.361), etc. before narrowing down to loneliness as the specific domain of interest.

5.       The authors state – “The public messaging on loneliness appeared to center around 12 topics based on topic modeling” Why is 12 the optimal number of topics here and not 13 or 14? Please present metrics from the topic modeling results to support this claim. 

There are a few minor sentence construction errors and grammatical errors.

Author Response

To begin, we thank the reviewer for taking the time and effort to review our manuscript. In addition to the changes below, we have also done a proofreading of the manuscript for grammar and language.

Comment 1: Rewrite the Methods section to clearly highlight how the Tweets were extracted. Was the Standard Search API used or the Advanced Search API? How long did the data collection take? How did the authors design their data collection to comply with the rate limits of accessing the Twitter API?

Reply 1: Thank you for the comments. We have rewritten the Methods section to provide further details and elaboration on the methods and techniques utilised. "Using Twitter as the social media platform of choice and the search term ‘#loneliness’, we extracted tweets that were posted in English from January 1, 2012 to September 1, 2022. The tweets were extracted from Twitter’s Application Programming Interface (API) platform, using an academic developer account, which allows downloads of all tweets (i.e. not sampling) of up to 10 million tweets per month."

Comment 2: Why was BERT Named Entity Recognition (NER) used instead of other topic modeling approaches such as LDA?

Reply 2: Thank you for the comment. We have now provided justification for our choice of BERTopic instead of other topic modelling approaches such as LDA in the Methods section, "BERTopic was chosen over other techniques such as Latent Dirichlet Allocation (LDA) as it is based on BERT embeddings, which are contextualized word representations learned through pre-training on large amounts of text data [19]. This means that BERTopic can take into account the context and meaning of words and provide more accurate and meaningful topic representations as compared to LDA, which uses a bag-of-words approach [19,20]." Supporting references that demonstrate a higher accuracy and performance for BERTopic as compared to LDA have been cited as well.

Comment 3: Please elaborate on how Twitter usernames were analyzed to deduce those were specific organizations. How accurate was this approach? How did the authors take into consideration that there may be bot accounts on Twitter which could have a name similar/identical to an organization?

Reply 3: Thank you for the comments. We have provided further elaboration on the use of BERT Named Entity Recognition (NER) in our Methods section, "only tweets by organisations were included in this study, as selected using BERT Named Entity Recognition (NER), which locates and identifies named entities, e.g., person names and organisations mentioned in unstructured free-text [17]. BERT NER has been trained using a pre-training and fine-tuning approach, and it uses a sequence labeling approach, where the model takes a sequence of tokens (words or subwords) as input and predicts a label for each token indicating whether it belongs to an entity or not, and if so, what type of entity it is. It is able to recognize four types of entities: location (LOC), organizations (ORG), person (PER) and Miscellaneous (MISC)." 

Nonetheless, we are unable to fully exclude the possibility of bots that have a name similar/identical to an organization as rightly mentioned by the reviewer. We have now added this to our discussion of study limitations, "Fourth, we cannot rule out the presence of bot accounts on Twitter that may have a name similar or identical to that of an organization, although research has shown that such bot accounts make up only a small proportion of users and tweets on the platform [34]. The sentiment expressed by these bot accounts is still relevant however, as it may influence what people are potentially exposed to on the platform."

Comment 4: The second paragraph in the Introduction section outlines just a couple of uses of social media networks such as Twitter in the context of COVID-19. Please elaborate on why people used Twitter in this context with supporting references – for instance: general information seeking and sharing (https://doi.org/10.3390/covid2080076), medical information dissemination (https://doi.org/10.1017/cem.2020.361), etc. before narrowing down to loneliness as the specific domain of interest.

Reply 4: Thank you for the comments and suggestions. We have incorporated the references provided and added in our Introduction section that, "Today, with the widespread use and availability of the internet and social media platforms like Twitter, people are increasingly turning to these platforms not only for general information seeking and sharing [7] and medical information dissemination [8] but also for emotional support."

Comment 5: The authors state – “The public messaging on loneliness appeared to center around 12 topics based on topic modeling” Why is 12 the optimal number of topics here and not 13 or 14? Please present metrics from the topic modeling results to support this claim. 

Reply 5: Thank you for the comment. Given the total number of tweets extracted, in BERTopic, the following hyperparameters were used: ngram of up to 3 words, minimum topic size of 0.5% of the total tweets, embedding model of “all-mpnet-base-v2”. This gave a total of 12 interpretable topics and a Miscellaneous topic for the remaining (unfitted) tweets.

Round 2

Reviewer 2 Report

The authors have revised their paper as per all my comments and feedback. I do not have any additional comments at this point. I recommend the publication of this paper in its current form.